# The sea level simulator v1.0: a model for integration of mean sea level change and sea level extremes into a joint probabilistic framework

Magnus Hieronymus[1]

[1]Swedish Meteorological and Hydrological Institute, Folkborgsvagen 17, Norrkoping, Sweden

**Correspondence:** Magnus Hieronymus (magnus.hieronymus@smhi.se)

**Abstract.** A statistical model called the sea level simulator v1.0 is introduced. The model integrates mean sea level change and sea level extremes into a joint probabilistic framework that is useful for coastal spatial planning. Given a user defined planning period, the model can estimate the flood risk as a function of height above the current mean sea level. These flood risk estimates are derived through Monte Carlo simulations of a very large amount of planning periods. The derived flood risk is contingent on user assigned probabilities for future greenhouse gas emission pathways, and the model is thus also useful for quantifying the dependence of flood risk on such pathways and their probabilities. Moreover, the simulator can quantify whether flood risk is dominated by sea level extremes or mean sea level rise and how this depends on the length of the planning period. The code, written in MatLab, is parallelized and lightweight enough that it can be run on an ordinary PC. The code is easily adaptable to include new locations, new mean sea level projections and similar model developments. The flood risk estimates derived from the simulator are well suited to tackle adaptation and decision problems. Applications for construction of coastal protection and land development in coastal areas have been demonstrated in the past. The paper gives an in-depth technical description of the model. Example simulations from a Swedish nuclear site are also given and the capabilities of the simulator are discussed. The main aim of the paper is to work as a technical reference for the first public release of the sea level simulator.

## 1 Introduction

Mean sea level change alters the probability of coastal flooding in communities around the world. The globally averaged mean sea level rose by about 20 centimetres during the period 1901-2018. However, much larger changes are projected for the current century in all future emission scenarios investigated in the Intergovernmental Panel on Climate Change Sixth Assessment Report (IPCC, AR6) (Fox-Kemper et al., 2021). Mean sea level change projected for the 21st century is, in fact, so sizeable that extreme sea levels that are expected to be reached on average only once in a century with the current mean sea level, could in many places be reached on a yearly basis even before the end of the current century (Oppenheimer et al., 2019; Hieronymus and Kalén, 2020). The probability of coastal flooding is thus expected to increase dramatically in many places, unless effective

protection is put in place, as a consequence of mean sea level rise. Even though mean sea level rise is the root cause of this problem, the time dependence of mean sea level change is rarely accurately accounted for in current coastal spatial planning (Hieronymus, 2021; Hieronymus and Kalén, 2022). Neither is, generally speaking, the uncertainty encompassed in probabilistic mean sea level projections. In fact, in a survey of 32 European countries by McEvoy et al. (2021), it was found that probabilistic mean sea level projections were only used in dedicated sea level planning in a single country and in non-dedicated planning in another three. Important planning decisions, such as how far above the current mean sea level new building and infrastructure can be erected, are instead typically based on arbitrary rules. In many places, the minimum distance above the current mean sea level where new buildings can be erected is determined by adding a high mean sea level projection most often for the year 2100 to a high return level for temporary sea level extremes (Arns et al., 2017). Sometimes an additional safety margin of arbitrary amount is also added to the two other components. The risk that buildings erected at such a distance from the current mean sea level could become flooded during their expected lifetime is presumably small. That is, as long as the level is derived from a sum of unlikely mean and extreme sea levels and the expected lifetime of the structure does not exceed the length of the mean sea level projection used. However, in real terms the risk is not quantified and therefore essentially unknown. This is of course a great obstacle, for example, for producing realistic cost benefit analyses that could underpin coastal spatial planning.

The sea level simulator framework introduced by Hieronymus (2021) and further developed by Hieronymus and Kalén (2022) rids the planner of much of the ambiguity inherent in these arbitrary levels. This is done by combining mean sea level projections and sea level extremes into a joint probabilistic framework. From this framework, flood risk can then be calculated as a function of height above the current mean sea level. This information can, in turn, be used to derive levels where new buildings can be erected that are based on the planners risk level preference. The modelled risk is contingent on probabilities given to different emission scenarios and the length of the planning period. The planning period in this context can be, for example, the lifetime of a structure, or the period over which the structures current value can be discounted to a suitably low level. Essentially, the sea level simulator is a tool for answering questions like: if a house is built $x$ m above the current mean sea level what is the risk that it will be flooded at least once during the next $y$ years. Answering that question requires no other data then that which is used to calculate the more arbitrary levels in use today. Essentially, what the sea level simulator does is to utilize these data in a better way and to formalize the underlying assumptions in a more rigorous manner. The latter point is of great importance. The assumptions we make about, for example, the expected life time of a structure or the probability of a given emission scenario coming to pass greatly affect the estimated risk of flooding. Another strength of the sea level simulator is that the influence such assumptions have on flood risk can be quantified in a straightforward manner. Thus, the simulator not only gives us probabilistic assessments of flood risk, it can also inform us about how these assessments depend on our basic assumptions about, for example, probabilities given to different future emission pathways or model projections of melt from the Antarctic ice-sheet.

The simulator uses two separate data sources in its calculations: mean sea level projections (Fox-Kemper et al., 2021; Hieronymus and Kalén, 2020; NASA, 2022) and time series of yearly sea level maximum relative to the mean sea level (Dan-

gendorf et al., 2016; Särkkä et al., 2017; Männikus et al., 2020). Any mean sea level projection can be used, but so far only those from the IPCC (Oppenheimer et al., 2019; Fox-Kemper et al., 2021) have been used in practise. Both the mean sea level projections and the time series of annual maximum sea level are fitted to continuous probability distribution functions. Sea level maxima relative to the current mean sea level can then simulated throughout planning periods by drawing yearly sea level maxima and mean sea level projections randomly from their respective distributions. The basic idea behind the simulator is to make vast amounts of such simulations from which frequencies of high sea levels in future periods can be determined.

The first study that utilized the simulator (Hieronymus, 2021) was a case study for Stockholm that showcased many of its potential applications for example to uncertainty quantifications, adaptation- and decision problems. The second study (Hieronymus and Kalén, 2022), focused on how the length of the planning period affects whether flood risk is dominated by mean sea level rise or sea level extremes, and how such knowledge could be utilized by coastal spatial planners. In the present paper, focus is put more on the technical aspects of the simulator. New updates to the simulator are also discussed. In particular the following.

- – implementation of parallel computing

- – implementation of new mean sea level scenarios

- – better uncertainty quantifications for sea level extremes

- – more realistic time dependence in the mean sea level projections

All of these improvements were absent in the simulator used by Hieronymus (2021) and Hieronymus and Kalén (2022). Moreover, the current paper is also intended to be the main technical reference for the first public release of the sea level simulator. This model has been given the version number 1.0 and is available free of charge, and warranty, to all through Hieronymus (2023).

## 2    Model description

### 2.1    An overview of the sea level simulator

The sea level simulator v1.0 is written in Matlab. The program consists of one main script and some support scripts used to set-up the model. It uses the statistics and parallel computing toolbox from MathWorks, as well as some routines from the free MatLab toolbox Cupid (https://github.com/milleratotago/Cupid). The Cupid routines needed by the simulator are distributed together with the simulator code. The parallel computing toolbox is only needed to speed up the computation, and running without it is essentially a matter of switching a *parfor* statement in the programs main loop into a *for* statement.

The program simulates a large amount of planning periods by randomly drawing time dependent mean sea level changes and yearly sea level maxima. The standard setting is $10^7$ such periods that last between 2021 and 2150. The time frame is chosen

so that it complies with the length of the mean sea level projections from Fox-Kemper et al. (2021). The probability of flooding

as a function of height above the current mean sea level is evaluated not just for the full period but also for all shorter planning

periods in ten year increments. That is, by modelling the planning period 2020-2150 we also get the probabilities for the 2020-2030 planning period, 2020-2040 planning period and so on, virtually free of any additional computational cost. Apart from information about the highest joint sea level (i.e. caused by both mean sea level rise and sea level extremes) experienced within the planning period, information is also stored about the heights of the mean sea level and the sea level extreme components

individually. The latter information is important to gauge potential impacts and to produce more informed risk assessments as was discussed by Hieronymus and Kalén (2022).

## 2.2   Step by step simulations and treatment of uncertainty

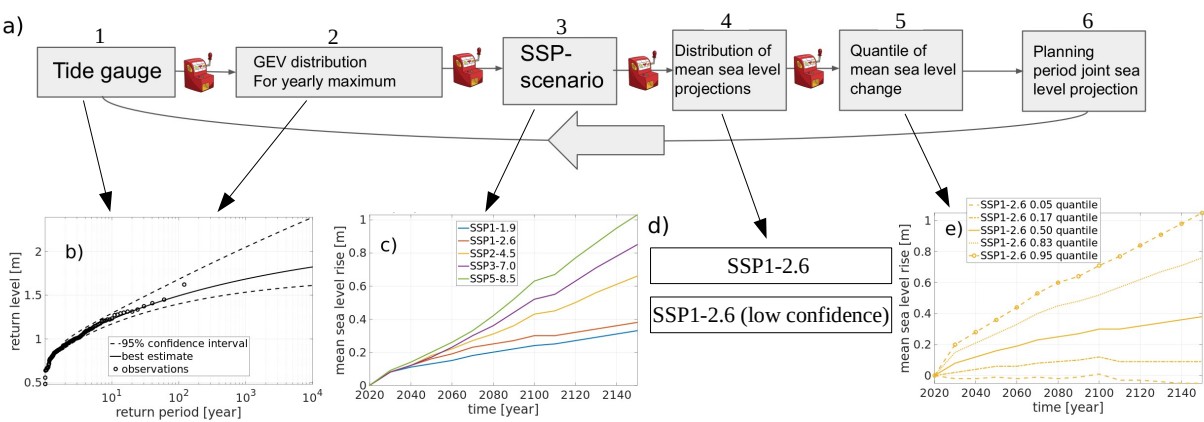

**Figure 1.** Schematic view of the sea level simulator with insets showing some of the data the simulator depends on. a) shows schematically how a planning period is simulated by going through a chain of modules from left to right. The large arrow below indicates that this cycle is repeated many time (i.e. that many planning periods are simulated). The one armed bandits signify that a stochastic process is involved in going from one module to the next. b) shows the GEV fit and the observed annual maxima, c) shows mean sea level projections for different SSPs, d) indicates that some SSPs have multiple mean sea level projections and e) shows some quantiles of the mean sea level projection for SSP1-2.6.

A schematic of the simulator is shown in Fig. 1a. The first piece of data needed to run a simulation is a time series of annual

sea level maxima from a tide gauge. This time series could, of course, also come from a numerical ocean model, a machine learning model or any other model that reliably captures high frequency sea level variability. What frequency and quality that is needed naturally depends on the location and the application, but most often hourly data should suffice. This annual maximum time series is then used to fit a Generalized Extreme Value (GEV) distribution (Coles, 2001), see Fig. 1b. This is done using the *gevfit* routine from Matlab's statistics toolbox. In earlier versions of the simulator (Hieronymus, 2021; Hieronymus and

Kalén, 2022), only the maximum likelihood estimate of the GEV parameters was used to characterize the tide gauge. That is,

in earlier versions there were no uncertainty in which GEV distribution the simulator used. Extreme sea level uncertainty was consequently only owing to the fact that each planning period had its yearly sea level maxima drawn randomly from the GEV distribution that was thought to best characterize the tide gauge in question. In this release, uncertainty is introduced also into the GEV parameters, which is signified by the one armed bandit connecting the first two modules of the schematic in Fig. 1a.

The GEV parameter uncertainty is turned on by setting the *ext_par_uncert* variable equal to one.

The uncertainty in the GEV parameters is modelled in the same way as maximum likelihood based confidence intervals on return levels are estimated with MatLab (Mathworks, 2020). Here we calculate the location, scale and shape parameters that define the GEV distributions that give rise to the upper and lower bounds on 1000 year return level confidence intervals. This

calculation is done for confidence levels between 0.01 and 0.99, using a resolution in confidence of 0.01. In total we get 199 sets of GEV parameters, which spans approximately the quantile range from 0.005 to 0.995 of plausible GEV distributions for the tide gauge. In each modelled planning period one of these parameter triples is chosen randomly, thus introducing stochasticity into the choice of GEV distribution used to model the annual sea level maxima at the site. In practise, a uniformly distributed random number from the interval $[0, 1]$ is drawn every planning period to give the quantile for the GEV parameters. The yearly

sea level maxima for the planning period are then drawn randomly from the randomly selected GEV distribution. The draw of the yearly maxima is signified by the second one armed bandit seen from the left in Fig. 1a.

Certain users might prefer to use peak over threshold rather than block maxima statistics, and consequently to model extremes with a Generalized Pareto (GP) instead of a GEV distribution. In v1.0 of the sea level simulator there is no such option

available. The main reason for this is that the GP approach requires more user defined parameters, such as a threshold and a separation time scale between events that should be long enough that the events can considered independent. Good guidance on how to choose these parameters is hard to give. The GEV approach has a similar parameter to the GP separation time scale, namely the block length. However, using a block length of one year is more or less standard practise (Arns et al., 2013), and using longer blocks is often impracticable owing to insufficient length of most sea level time series (Hieronymus and Hi-

eronymus, 2023). Nevertheless, for a user wanting to use the GP approach only minor code changes are needed, and Matlab's statistics toolbox contains the necessary *gpfit* and *gprnd* routines.

A further issue worthy of note regarding the extreme sea level distributions used by the simulator are that these are unaffected by climate change and time. That is, the GEV distributions used are independent of both time and Shared Socioeconomic

Pathway (SSP). This is simply because of lack of knowledge about how the GEV parameters might change through time under given SSPs. However, if such knowledge was available it would be easy to include climate change induced trends in annual sea level maximum as a perturbation to the mean sea level projections. This can be done without any code changes to the simulator. How the mean sea level projections are made is discussed further down in this section.

In the next module a SSP is chosen randomly. Five different SSP-radiative forcing combos are available from Fox-Kemper et al. (2021); NASA (2022): SSP1-1.9, SSP1-2.6, SSP2-4.5, SSP3-7.0 and SSP5-8.5 (Fig. 1c). The numbers after the dash indicate the radiative forcing in $\text{Wm}^{-2}$ in the year 2100 compared to a preindustrial baseline. Each SSP-radiative forcing combo is assigned a probability, $(p_1, ..., p_5)$, of coming to pass, and these probabilities are chosen so that $\sum_1^5 p = 1$. The next step is to pick a distribution for the projected mean sea level. Here only scenarios SSP1-2.6 and SSP5-8.5 has more than one option (Fig. 1d). These two scenarios have also *low confidence* projections, where the contribution from the Greenland and Antarctic ice-sheets to sea level rise is taken from some of the highest projections in the published scientific literature (Bamber et al., 2019; DeConto et al., 2021). The uncertainty introduced by having more that one mean sea level distribution per emission scenario is similar in nature to the uncertainty introduced to the GEV parameters. That is, in both cases there is uncertainty both in the underlying distributions and in the random numbers drawn from the chosen distributions. The contributions to sea level rise from other components such as thermosteric expansion and melting glaciers are the same in the *low confidence* as in the main projections. In practise the lottery over climate scenario and sea level projection is combined into one joint lottery in the code. This is done because only two climate scenarios have multiple projections and such an implementation is a little faster. That is, the lotteries in the third and forth modules in Fig. 1 are implemented in the code as a single lottery over the available distributions of projected future mean sea levels. Just as for the GEV parameters, this random process is modelled by drawing a uniformly distributed random number from the interval $[0, 1]$.

This random number maps to a mean sea level projection through the user defined probability range for the projections. Note that neither the sea level projections nor the SSPs have been attributed probabilities by their makers. However, for the SSPs, at least, some estimates of suitable probabilities have been derived using integrated assessment models (see e.g. Capellán-Pérez et al. (2016) and Huard et al. (2022)). An example probability range is shown in Tab. 1. The random number used for drawing the mean sea level projection is independent of that used to pick the GEV distribution.

**Table 1.** Probabilities given to the different mean sea level projections and the probability range in which the different projection are applied.

| mean sea level distribution | probability | probability range |
| --- | --- | --- |
| SSP1-1.9 | 0.05 | [0, 0.05] |
| SSP1-2.6 | 0.155 | (0.05, 0.2050] |
| SSP1-2.6 (low confidence) | 0.01 | (0.2050, 0.2150] |
| SSP2-4.5 | 0.5 | (0.2150, 0.7150] |
| SSP3-7.0 | 0.22 | (0.7150, 0.9350] |
| SSP5-8.5 | 0.064 | (0.9350, 0.999] |
| SSP5-8.5 (low confidence) | 0.001 | (0.999, 1] |

The distributions of mean sea level projections are available every ten years (Fox-Kemper et al., 2021; NASA, 2022). Thus, for each mean sea level projection there is one distribution for 2030, one for 2040 and so on. To get consistent mean sea level projections, a quantile is chosen randomly for each modelled planning period, and this quantile of the chosen mean sea level projection is extracted from the mean sea level distributions from each time step. Linear interpolation is then done to get a mean sea level projection with yearly resolution (Fig. 1e). Having mean sea level projections for every ten years is another improvement over the earlier versions of the simulator. Those versions used only one distribution for the year 2100 per emission scenario and uncertainty was set to grow linearly with time from zero at the start of the planning period to its end value in 2100 (Hieronymus, 2021; Hieronymus and Kalén, 2022). Similarly to the earlier stochastic processes the mean sea level quantile is also chosen by drawing a uniformly distributed random number from the interval $[0, 1]$. This random number is independent from those determining the GEV quantile and mean sea level projection used.

The IPCC mean sea level distributions are discreet. For use with the simulator they are therefore approximated by continuous skewnormal distributions. The three parameters defining the skewnormal distributions are chosen so that the sum of the squared differences between the continuous skewnormal distribution and the discreet IPCC distribution is minimized at the 5th, 17th, 50th, 83rd and 95th percentiles. These difference are very small, typically within a cm, for all mean sea level distributions except those for SSP5-8.5 *low confidence* for years 2100-2150, where we cannot get a good approximation for both the 83rd and 95th percentiles at the same time with the skewnormal distribution. In fact, many different continuous distributions (e.g. Gaussian, exponential, Weibull and exponentially modified Gaussian) have been tested without finding a good fit. For these distributions the difference between 50th percentile and the 83rd percentile is very much larger than that between the 83rd and the 95th percentile, suggesting that SSP5-8.5 *low confidence* for these years is likely bimodal. In these cases, we have opted to minimize the sum squared difference at only the 5th, 50th and 95th percentiles to have a good accuracy at the highest percentiles. The fitting of the skewnormal distribution parameters is done using the included script *meanseadists.m*. This script can easily be edited for use with different mean sea level projections. It is also worth mentioning that mean sea level projections are site specific and the projections used as an example here are for the area on the Swedish west coast where the nuclear power plant Ringhals is situated. When setting up the simulator for a new location it is thus important to check how well the continuous distributions, whose parameters are estimated using *meanseadists.m*, agree with the discreet originals. This is also diagnosed in the *meanseadists.m* routine. Moreover, the Cupid toolbox offers many alternatives to the skewnormal distribution that one can easily adapt the code to use. For Swedish conditions the skewnormal distribution has proven to give good fits, but this could conceivably be location dependent.

The sixth module in the diagram adds the mean sea level projection for the planning period to the annual sea level maxima modelled for the period. The resulting time series contains the planning periods annual maxima referenced to the current mean sea level. It is from this time series that the planning period sea level maximum and its mean sea level and extreme sea level components are extracted. These variables are saved in form of discreet probability density functions (PDFs). The grid resolution of these PDFs is set by the parameter *nr_res*, which has a standard value of 500. This means, that 500 equidistant

grid points are used to resolve the planning periods sea level maximum. For each planning period, one is added to the grid point whose value is closest to the simulated sea level maximum. When all iterations are done, the result is normalized by the

200 amount of planning periods modelled to get PDFs from the derived histograms. The loop between the different modules in Fig. 1a is repeated for the desired amount of planning periods. In the examples that follow, $10^7$ planning periods are modelled in each experiment. The statistics produced by the simulator are thus planning period probabilities rather than the commonly used yearly probabilities. For example, a relative frequency of $10^{-4}$ means that one in ten thousand planning periods contain a sea level of this height. The term planning period probability is therefore used here instead of the more commonly used

terms yearly probability and return period. The reason for this is that the yearly probability or return period of a given sea level changes throughout the planning period, while the planning period probability is stationary.

## 2.3   Parallel computing performance

As was mentioned in the introduction, multiple planning periods are simulated in parallel. The scaling of runtime vs amount of

210 cores, shown in Fig. 2, is very close to the ideal. Thus, without parallel computing the program will be much slower. The close to ideal scaling holds only for physical cores, at least on the test machine. Using hypertreading to run on up to four virtual cores together with the four physical cores only affected the runtime in a minor way. There is, however, a good workaround for those who do not have the parallel toolbox, but require a large amount of planning periods to be simulated. The problem of simulating planning periods is, in fact, embarrassingly parallel. Therefore, the program can be run on multiple computers and

215 the results can be averaged afterwards.

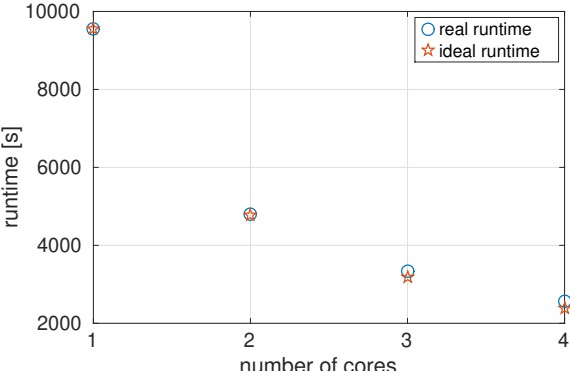

**Figure 2.** Runtime plotted against amount of cores for the sea level simulator v1.0. In the experiment shown $10^5$ planning periods are simulated. The experiment is done on a laptop using a Intel Core i5-8265U CPU @ 1.60GHz x 8. The ideal runtime is computed as the runtime on one core divided by the number of cores used.

## 3 Example simulations

In the examples that follow, the sea level simulator has been set-up to model annual maximum water levels at the Swedish nuclear power station Ringhals. Ringhals is situated in Varberg municipality on the Swedish west coast. The sea outside Ringhals is called Kattegat. It is a shallow sea situated between the Baltic Sea and Skagerak to the south and north, and Sweden and Denmark to the east and west. Kattegat has weak tides and strong stratification. The strong stratification is a consequence of low saline Baltic Sea water meeting more saline North Sea water, yielding a hydrography well described by a two layer system (Leppäranta and Myrberg, 2009; Lehmann et al., 2022). The main examples shown are for a model run with scenario probabilities according to Tab. 1. One run is also done where the *low confidence* projections are given a probability of occurrence equal to zero. In this case, their former probabilities are instead added to the main SSP2-2.6 and SSP5-8.5 projections so that the probabilities for all projections sum to one.

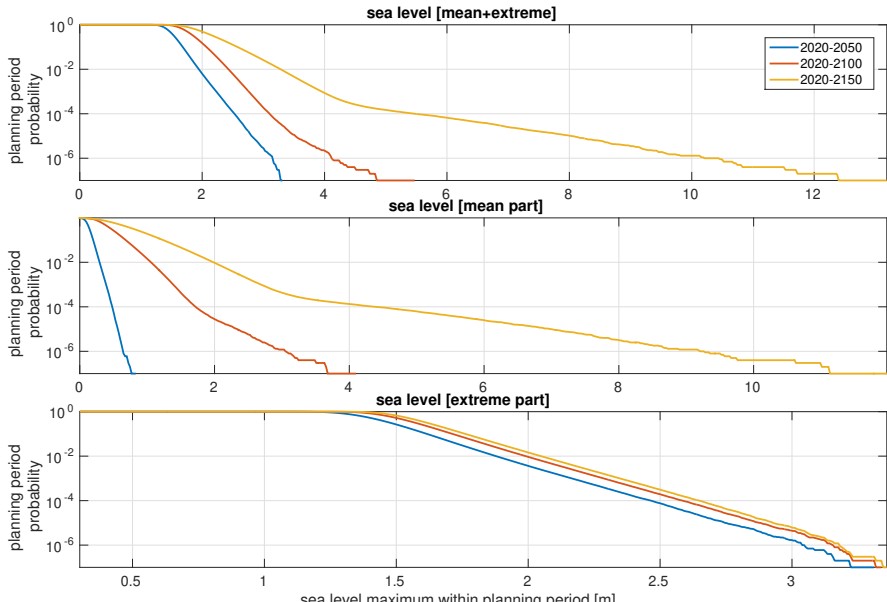

**Figure 3.** Planing period probability of sea level maximum for three different lengths of the planning period. The top panel shows the joint sea level maximum, the middle the maximum mean sea level and the lower panel the maximum extreme sea level. The figure is derived using the mean sea level projection probabilities given in Tab. 1.

Figure 3 shows the cumulative distribution of the sea level maximum relative to the current mean sea level for three different lengths of the planning period. It is readily evident from the figure, that in the two longest planning periods it is the mean sea level change that gives rise to the highest sea levels, while temporary sea level extremes are responsible for the highest sea levels in the shortest planning period. This is consistent with earlier work at other Swedish locations, using an older version of the sea level simulator (Hieronymus, 2021; Hieronymus and Kalén, 2022). The dominance of the mean sea level contribution over

that from the extremes is here exacerbated compared to earlier work because of the inclusion of the SSP5-8.5 *low confidence* projection. In this simulation, that scenario is given a probability of occurrence equal to $10^{-3}$, and both the mean and the mean + extreme panel of the plot are dominated by this scenario at frequencies lower than approximately $10^{-3}$.

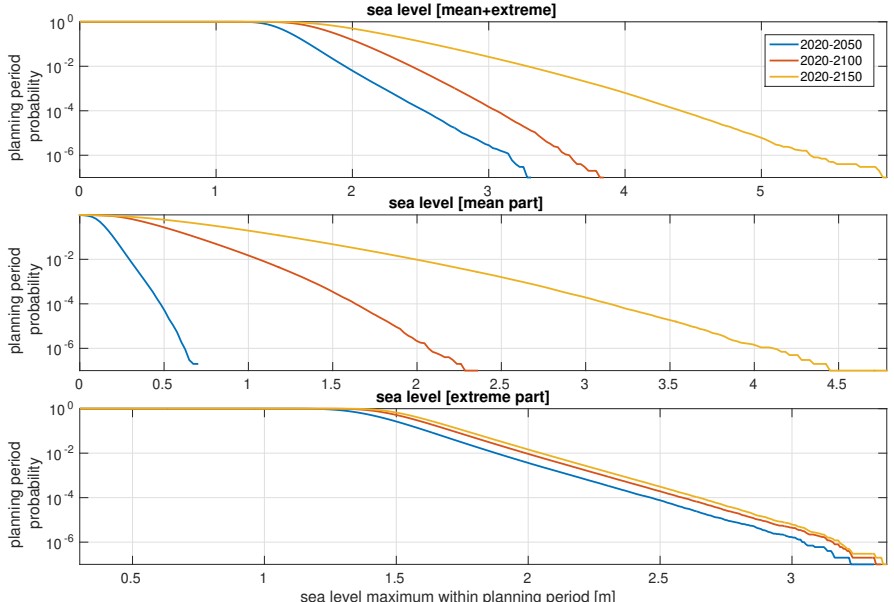

**Figure 4.** Same as Fig. 3, but with zero probability given to the *low confidence* versions of SSP2-2.6 and SSP5-8.5.

Figure 4 shows a simulation where the probabilities given to the *low confidence* projections are set to zero. Here, there is no regime shift for frequencies smaller than $10^{-3}$, and the low frequencies are more of a natural continuation of the higher ones. For future developments, it could be useful to incorporate other high-end mean sea level projections that are not quite as disjoint from the main projections as SSP5-8.5 (*low confidence*). A notable example of such projections, are the recently published high-end projections by van de Wal et al. (2022). These projections are designed through a community effort and aim to give physically plausible high-end projections for two different warming levels. It should be noted, however, that the range of sea levels projected without the *low confidence* projections is, in fact, considerably larger than the high-end projection for 2100 by van de Wal et al. (2022), which gives 1.6 m of globally average sea level rise for a global warming of five degrees. One could thus argue that the high-end, or at least the physically plausible high-end is already included in the main SSP5-8.5 projection. That is, even without separate high-end projections.

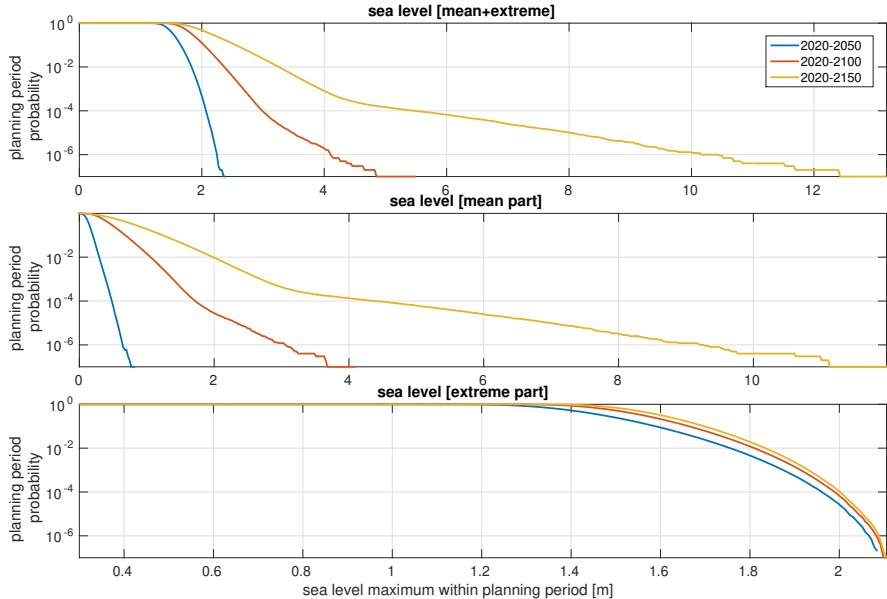

**Figure 5.** Same as Fig. 3, but with all extremes drawn from the same GEV distribution. That is, all simulated planning periods have their extremes drawn from a GEV distribution that uses the maximum likelihood parameter estimates inferred from the tide gauge data.

The influence of the GEV parameter uncertainty on the joint sea level maximum is sizeable in short planning periods, but relatively insignificant in long ones. Figure 5 is the same as Fig. 3, but here all extremes are drawn from the same GEV dis-
250 tribution. That is, the first lottery in Fig. 1 is cancelled, and we assume that the maximum likelihood GEV parameters are the true ones. Here we find that even though there is a very significant change in the extreme sea level, the change in the joint sea level is rather modest in the longer planning periods. The reason for this behaviour is, of course, that the range of the modelled mean sea levels becomes much larger than the range of modelled extremes in long planning periods. However, how long a planning period has to be for the extreme contribution to become small compared to the mean contribution is a function of
255 the probabilities given to the emission scenarios, and the length of the time series of yearly maxima used to infer the GEV parameters. It also seems prudent to point out that the extreme range simulated with the GEV parameter uncertainty turned on is not necessarily physically plausible. This range depends on the length of tide-gauge time series, and not on understanding of the local oceanographic conditions. For locations, where only short time series are available, it could thus be useful to use data also from neighbouring tide gauges (Calafat and Marcos, 2020; Räty et al., 2022) or from numerical ocean models (Särkkä
et al., 2017; Hieronymus and Hieronymus, 2023) to better constrain a plausible range of GEV parameters.

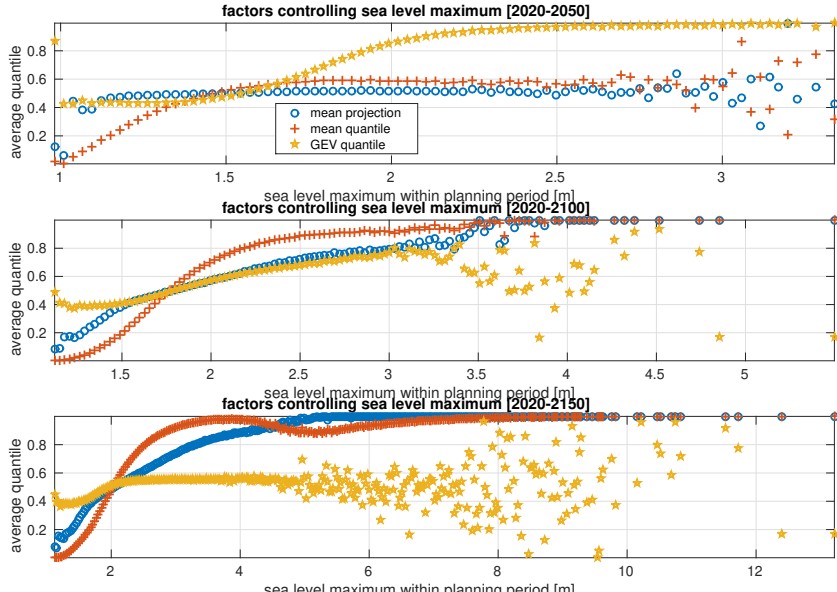

**Figure 6.** The average outcome of the stochastic processes depicted in Fig. 1 as a function of planning period sea level maximum. The three panels show different lengths of the planning period. Mean projection indicates which projection that gave rise to the sea level maximum on the x-axis. The mapping of average quantile to mean sea level projection uses the probability range given in Tab. 1. The mean quantile shows the quantile of the used sea level projection and GEV quantile shows the quantile for the GEV parameters. Higher average quantiles mean higher sea levels in all cases.

The simulator can also be used to infer how the outcome of the different stochastic processes depicted in Fig. 1 affect the planning periods sea level maximum. This is illustrated in Fig. 6, where the average quantiles of the uniformly distributed random numbers used to model the different uncertainties are shown as a function of planning period sea level maximum. The mean projection average quantile can be mapped to the specific sea level projection used through the probability range in Tab. 1. GEV quantiles refer to the modelled uncertainty in the GEV parameters, and higher quantiles give GEV distributions with higher extremes. The GEV quantile is thus not a direct measure of the height of the extremes, but of the propensity of the GEV distribution to give high extremes. The figure, shows in a quantitative way what we could already deduce from Fig. 3. Namely, that in the shortest planing period the highest sea levels are almost independent of mean sea level change, but are strongly dependent GEV parameter uncertainty. In the two longer planning period the situation is reversed and the highest sea levels all occur under a high quantile of the SSP5-8.5 *low confidence* projection. Moreover, it is also clear that the highest sea level in the longer periods occur independently of the GEV quantile in the two longer planning periods.

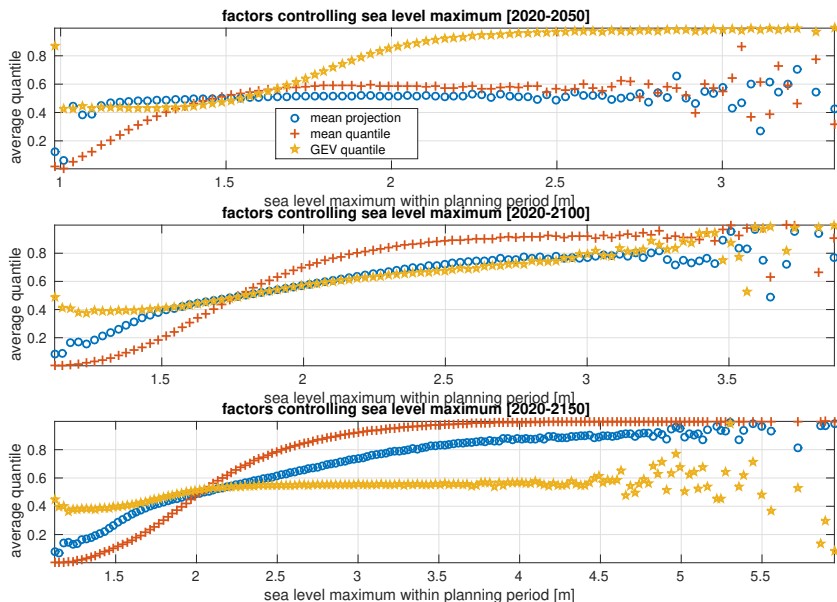

**Figure 7.** Same as Fig. 6, but with zero probability given to the *low confidence* versions of SSP2-2.6 and SSP5-8.5.

Figure 7 shows the same diagnostics as Fig. 6, but here we have given the *low confidence* projections for SSP1-2.6 and
SSP5-8.5 zero probability of occurrence. Qualitatively the behaviour is the same as when the very high-end mean sea level
projection are included. However, the highest sea level are of course significantly lower without the SSP5-8.5 *low confidence*
projection. The effect of excluding the SSP1-2.6 *low confidence* projection is, however, small. Regardless of whether the high-
end projections are included, it is clear that the switch from high sea levels being dependent on high GEV quantiles to being
dependent on high mean sea level projections and quantiles occurs within this century at Ringhals with the given set of emis-
sion scenario probabilities.

The sea level simulator can also output directly the respective contributions from sea level extremes and mean sea level
rise to joint sea level events. Such a diagnostic is shown in Fig. 8. The upper panels show the relative density of the extreme
and mean sea level contributions to the joint sea level maxima. The lower panels show the mean of these distributions for
three different planning periods. In Fig. 9 the same diagnostics are shown for the case with no GEV parameter uncertainty.
Both figures tell the same story about how the highest joint sea levels go from being dominated by the extreme contribution in
short planning periods to being dominated by mean sea level rise in long planning periods. However, the quantification of the
respective magnitudes of the individual components give some added value. Mean sea level rise and sea level extremes occur
on very different time scales, and it is not necessarily always their sum that is the only concern. Note that, extreme sea levels
occur under severe storms and are likely to be picked up by numerical weather forecast systems, so that warnings can be issued
a day or two in advance. In contrast, multimeter mean sea level rise would, if it were to occur, be apparent decades in advance.

The set of actions that can be implemented to adapt to these two different treats are thus very different. Knowledge of their respective magnitude can thus help inform decision makers on the best options available.

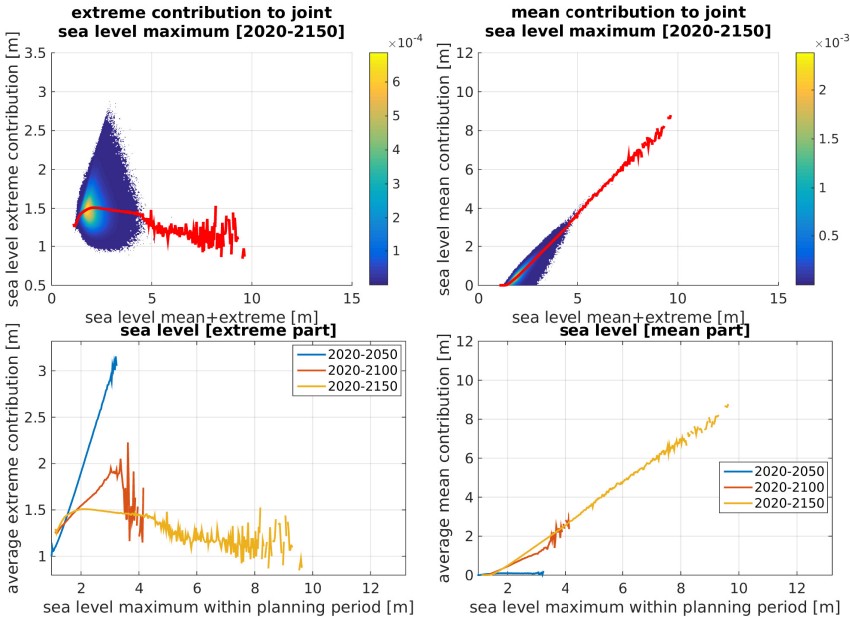

**Figure 8.** Mean sea level and extreme sea level contributions to joint sea level maximum. Upper panels show relative density [unitless] of the extreme and mean sea level contributions to the joint sea levels. The lower panels show average extreme and mean sea level contributions for different lengths of the planning period. Similarly, the red lines in the upper panels show the average extreme and mean sea level contributions for the 2020-2150 planning period.

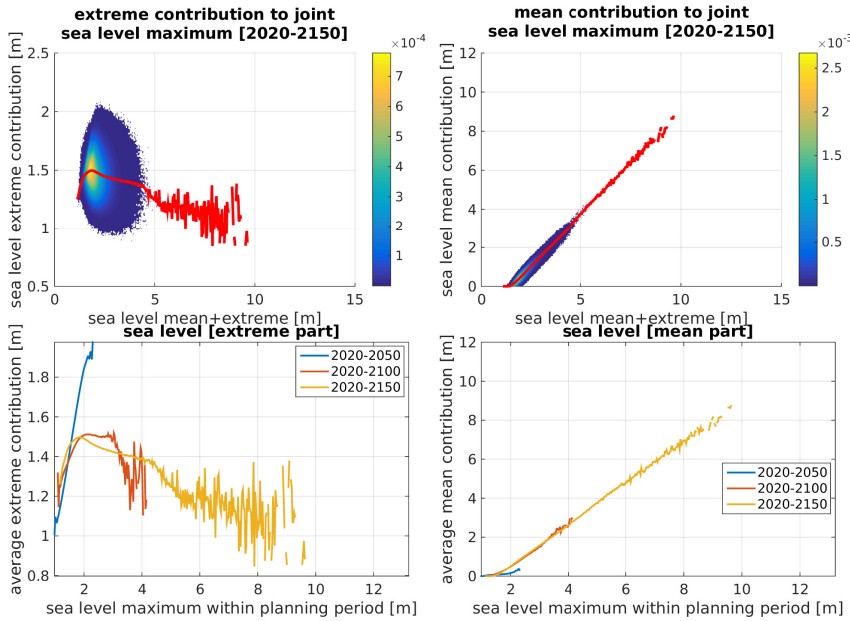

**Figure 9.** Same as Fig. 8, but with all extremes drawn from the same GEV distribution. That is, all simulated planning periods have their extremes drawn from a GEV distribution that uses the maximum likelihood parameter estimates inferred from the tide gauge data.

## 4  Conclusions

The modelling framework incorporated into the sea level simulator v1.0 has been presented in detail, and example simulations for a Swedish nuclear site have been discussed. Earlier versions of the sea level simulator have been used in scientific publications (Hieronymus, 2021; Hieronymus and Kalén, 2022). However, this publication marks the first public release of the source code and is aimed at being a technical reference publication, while the earlier two papers had different foci. Moreover, the sea level simulator v1.0 features several major updates that were not available in earlier code versions. Most notably, in terms of new scientific content we have the new mean sea level projections from (Fox-Kemper et al., 2021) and the implementation of the GEV parameter uncertainty. In technical terms the parallelization of the code is the most notable new feature. The code is not excessively numerically expensive to run. Most of the examples used in this presentation were run on a laptop with an Intel Core i5-8265U CPU @ 1.60GHz x 8 processor. The framework should thus be possible to implement even for small municipalities that don't have time on computational clusters as an expense in their yearly budgets.

The code is easily adaptable to new locations and uses widely available input data of the same kind that is used in more traditional methods of sea level planning. Essentially, what is needed to run simulations are a time series of yearly sea level maxima and at least one mean sea level projection. Apart from the obvious usage for creating decision support, the sea level simulator is also extremely well equipped for making uncertainty quantifications. A feature that has been further illustrated in

a number of examples by Hieronymus (2021) and Hieronymus and Kalén (2022). Further possible applications are to embed the sea level simulator into adaptation and decision problems. This was exemplified by Hieronymus (2021) who showed that conditioning adaptation measures on mean sea level rise would be an effective strategy for Stockholm. In the same paper, it was also illustrated how the simulator could be used to estimate whether it would be profitable or not to develop land depending on its height above the current mean sea level.

The list of possible new applications is extensive. An obvious but yet unexplored possibility would be to use the simulator to estimate future flood damage costs using information on the height above sea level and value of existing infrastructure. Moreover, the statistical framework could be used to model other hazards where short weather related events are superimposed on long term climate related trends. Heatwaves would be one such possibility.

In the current implementation, the vast majority of the runtime is spent making the mean sea level projections for the planning period. The most time consuming part is to find the inverse of the CDF of the mean sea level projections, which gives the mean sea level projections for the desired quantile. If this part could be sped up it would lead to significant decreases in the overall runtime. Nevertheless, in its current form the simulator is still fast enough that it can be run on an ordinary PC, and speed is thus mostly an issue for users who wishes to run very many simulations.

The GEV distributions used to model yearly sea level maxima are taken to be independent of time and SSP, which is an obvious caveat. In the Swedish context, this is likely a fair approximation given that Hieronymus and Hieronymus (2023) found trends in yearly sea level maxima to be largely independent of representative concentration pathway (RCP), in a large ensemble of downscaled climate projections. However, in most locations there is simply no data available on which SSP-based trends in yearly sea level maxima can be based. Otherwise, it would be straightforward to add the relevant trend to the already time dependent mean sea level projections. Although such an implementation would make the separation into mean and extreme sea level components less direct.

Lastly, it seems prudent to mention that both mean sea level projections (Jevrejeva et al., 2018; Horton et al., 2018; Le Bars, 2018; Hieronymus, 2020) and extreme sea level estimates (Suursaar and Sooäär, 2007; Dangendorf et al., 2016; Wahl et al., 2017) come with very large uncertainties . Moreover, especially mean sea level projections handle inherently subjective probabilities, can be widely diverging and change considerably from time to time. The same is also true about future emission scenario probabilities. The simulator framework does not help constrain these uncertainties. The fidelity of the simulations is thus a function of the fidelity of the underlying data. However, the simulator is extremely useful to pinpoint which uncertainties that have the largest influence on flood risk. It may thus prove useful to direct research efforts into attempts at narrowing the uncertainty ranges that are the most important for flood risk.

*Code and data availability.* The current version of model is available from the project website: https://github.com/m-hieronymus/the_sea_level_simulator under the MIT licence. The exact version of the model used to produce the results used in this paper is archived on Zenodo (Hieronymus, 2023), as are input data and scripts to run the model and produce the plots for all the simulations presented in this paper (Hieronymus, 2023).

*Author contributions.* Magnus Hieronymus wrote the manuscript, invented the simulator and performed the experiments

*Competing interests.* I declare no competing interests

*Acknowledgements.* The work with the sea level simulator v1.0 is partly supported by the project Nuclear pOwer And long tail flood risK (NOAK), which is financed by the Swedish radiation safety authority. I would also like to thank two anonymous reviewers for their helpful comments.

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
