# Peer review of "The sea level simulator v1.0: a model for integration of mean sea level change and sea level extremes into a joint probabilistic framework"

_Geoscientific Model Development, 2022_

## Referee Comment (RC2)

Review Hieronymous. The sea level simulator

This paper describes a simple statistical tool that combines trends in mean sea level and extremes for planning purposes.

The paper is reasonably easy to access but requires here and there a bit more context to ensure proper use of the simulator. Below some suggestions for change which are all easy to accommodate in my view.

Line 73. A more conceptual introduction to the simulator is needed. You can not expect the reader to be familiar with Hieronymous 2021 and Hieronymous 2023.

Line 75. From the blue sky you start to discuss the parallelization aspects of the simulator. It is good to mention them somewhere in the paper, but not right at the start. In the end, being used to using large models, I don't think the parallelization is critical for the users to decide whether they use the tool or not. So please move backward in the paper or to an appendix.

Line 86 define what a planning period is.

Line 95 Explain in more detail how mean sea level is combined with extreme information this can be done in various ways. A paragraph of discussion is needed. In the literature a lot of studies on extreme sea level have long discussion whether a GEV or Pareto distribution should be used and whether joined variability issues should be treated and which peak threshold are to be used and what declustering of the data. This is relevant to discuss as for many applications the GEV is maybe not the best strategy, so make the reader aware or better, but demanding expand the tool in this direction.

Line 98 please rephrase "is goes"

Line 115. If I understand correctly the GEV itself is not changing over time due to the climate change itself. Probably the only thing you can do, but likely to be incorrect as well. Most climate variables change in the mean and in the pdf. So you at least have to create awareness among your readers of this point.

Line 141 you have to mention that SSP scenarios don't have a probability in their definition, but you implicitly use a probability that they are equally likely if I understand correctly, this is not how they are defined

Line 164 rephrase ones?

Line 195 That seems a trivial discussion. It is always correct in the extremes. At t=0 the trend is zero so extremes rule. If t goes to infinity SLR goes to extreme large values at least for high scenarios and the trend will rule. Explain this better in forehand and than show your examples where results are a mixture and both components might be important.

Figure 5 and others add units ot the vertical scale (1/yr)?

---

## Author Comment (AC1)

**Response to the questions of reviewer one**

First of all I would like thank the referee for his/her thoughtful comments, time and interest in the manuscript. I have put the referees comments in italics and my answers are given in plain text.

*Hieronymus presents the sea level simulator v1.0 following the first official public release of the source code. The sea level simulator aims at providing the joint probability of mean sea level rise and sea level extremes which is crucial for coastal planning and future adaptation. This tool could be particularly useful for engineering purpose and decision-making.*

*The manuscript acts as a technical reference publication as stated by the author. The framework behind the sea level simulator is not new and studies using previous versions of it have previously been published (Hieronymus, 2021; Hieronymus and Kalén, 2022). This version offers some updates such as new mean sea level projections (based on Fox-Kemper et al., 2021) and the implementation of uncertainty in the GEV parameters for the extreme sea levels.*

*The second part of the paper focus on a particular site and shows how the model behaves based on different parameter choices. More importantly, the manuscript discusses what one can infer from the model outputs. For example, which physical process is the dominant at different time horizons; this has direct impact when designing new structures and their planning periods.*

*The code is written in Matlab and released under MIT licence. Having no access to a Matlab licence, I have not tested the code directly, but the Gitlab is easily accessible. Users could potentially adapt and change the routines thanks to the use of Matlab. The different functions seem relatively straightforward to read though some users might find difficult to follow due to the lack of comments and descriptions inside each Matlab file.*

*GMD seems to be a good fit for the manuscript considering the author aim to publish a reference technical publication on the sea level simulator. Considering the previous two peer-reviewed papers on the framework, the comments below are mostly minor but would make the paper easier to digest by readers.*

*General Comments:*

*A significant part of the paper looks at what one can infer from the simulator (is the extreme level or SLR level the key driver, at which time horizon, etc), but it's not mentioned in the abstract. I think it is a very interesting capability and a key aspect of the simulator as it will be part of the decision-process a user might look at. It should be a bit more highlighted.*

Clearly this is an omission on my part. I will make sure to include a better

description of the simulator's capabilities in the revised abstract.

*As this paper should be viewed as a technical documentation, I think it would highly benefit from an extended Figure 2 where each step is associated with a sub-figure showing the raw time serie data, the GEV fits for extreme levels, the SLR scenarios, the sampling etc. It would really guide the reader. (I would also use more the word components / modules instead of nodes).*

I will include such a figure in the revised manuscript, and use modules instead of nodes.

*"An example probability range is shown in Tab. 1" line 138; it might be good to have some guidelines for a user in how to choose these probabilities. As shown in the second part of the paper, they do have an impact.*

I will add a discussion and some references about how one can think about these probabilities.

*I wonder if having another site with a potentially different behaviours or with different probability for the scenarios could be useful for a reader.*

This is the only reviewer suggestion that I object to. The primary reason for this is that I believe it would be very hard to keep the article as a technical reference publication. I think it would almost certainly drift towards discussing interesting questions about how regionally varying oceanographic conditions affect flood risk. This is, of course, an interesting topic, but one that in my opinion deserves its own article. A secondary reason is that in the Swedish context this has already been done in Hieronymus & Kalén (2022), where six sites with very long observational records were simulated.

*More extensive comments in the source would be beneficial to the users (mainly in emulator_ringhals_par.m, example_plot.m which seems to be the drivers); the associated set of functions are much more commented.*

I will update both these scripts with additional comments.

*The figures could be improved for better readability: none have a grid, the labels, ticks, and legends have generally small font size. I also believe showing the y-axis in term of return period would be more intuitive to a user / reader. This could be done on the right side with a twin axis for an example. The return period is stated in the text but shown in figure would be appreciated: "For example, a frequency of 104 means that one in ten thousand planning periods contain a sea level of this height."*

I will update all figures in the revised version following the reviewers suggestions.

*Specific Comments:*

*Line 80: "very close to the ideal"; does the model keep its scaling skills for 16, 32 cores?*

My test computer only has four cores, but the answer is nonetheless almost certainly yes. That is, as long as you make a large amount of simulations so that each worker can get a significant workload. As mentioned in the manuscript the problem of running these simulations is, in fact, embarrassingly parallel so simulations can be run simultaneously on different computers and averaged afterwards. The overhead related to initialization, loading data and such is really small when millions of simulations are run.

*Figure 2 caption: "is goes"; remove "is"*

this will be fixed.

*Line 100: "that reliably captures high frequency sea level variability"; maybe precise for the reader what "high-frequency" means so they do plug the right data to the model.*

I will add a sentence about this.

*Line 127-128: "These two scenarios have also low confidence projections, where the contribution from the Greenland and Antarctic ice-sheets to sea level rise is taken from some of the highest projections in the published scientific literature." A citation would be helpful here.*

Citations will be added.

*Line 145-147: I find this sentence not very clear.*

I will rewrite it.

*Line 157: it could be useful to precise which other distributions have been tested to know what does and doesn't work.*

I expect that what works might be both site and scenario specific, so I am not sure one can generalize all that much from my experience. Especially since Swedish mean sea level projection are quite atypical. I will add some more details regardless.

*Line 161: "This script is easily edited for use with different mean sea level projections"; can easily be edited ?*

The sentence will be rewritten according to the suggestion.

*Line 179: "For example, a frequency of $10^{-4}$ mean that one in ten thousand planning periods contain a sea level of this height." Replace mean by means*

this will be fixed.

*Line 183: maybe more "sea" than "ocean".*

Indeed.

*Line 186: It would be good to reference here some previous studies when discussing the stratification in the North Sea / Baltic Sea regions. A recent study I found that could help finding the most appropriate references https://esd.copernicus.org/articles/13/373/2022/*

I will add some references.

*Figure 8: what is the colorbar for? or is it lost in the middle of the dark dots.*

I forgot to turn on the shading. The figure will be redrawn of course.

*Figure 8: I might misread the figure but why the extreme levels impact reduces with longer sea level maximum for different length of planning periods? Why not constant.*

You are reading it correctly. The reason is simply that when the planning period is short the joint sea level (i.e. mean+extreme) is almost equal to the extreme contribution. The mean sea level simply does not change that much in any projection between 2020 and 2050 and the only way to get say a 3 m joint sea level event is to have an extreme sea level of almost 3 m (a very unlikely extreme at Ringhals). Conversely, when your planning period is very long, mean sea level change can be sizeable in many different projections and percentiles. Therefore, there are many more combinations of mean and extreme in the 2020-2150 planning period that gives a 3 m joint sea level that contain a large mean sea level contribution and a more average extreme contribution than the opposite.

---

## Author Comment (AC2)

**Response to the questions of reviewer two**

First of all I would like thank the referee for his/her thoughtful comments, time and interest in the manuscript. I have put the referees comments in italics and my answers are given in plain text.

*Review Hieronymus. The sea level simulator*

*This paper describes a simple statistical tool that combines trends in mean sea level and extremes for planning purposes. The paper is reasonably easy to access but requires here and there a bit more context to ensure proper use of the simulator. Below some suggestions for change which are all easy to accommodate in my view.*

*Line 73. A more conceptual introduction to the simulator is needed. You can not expect the reader to be familiar with Hieronymous 2021 and Hieronymous 2023.*

I will add a more conceptual introduction through an extension of the paragraph starting on L38.

*Line 75. From the blue sky you start to discuss the parallelization aspects of the simulator. It is good to mention them somewhere in the paper, but not right at the start. In the end, being used to using large models, I don't think the parallelization is critical for the users to decide whether they use the tool or not. So please move backward in the paper or to an appendix.*

I will move the paralellization part further back toward the end of Sect. 2.

*Line 86 define what a planning period is.*

A definition is given on L42.

*Line 95 Explain in more detail how mean sea level is combined with extreme information this can be done in various ways. A paragraph of discussion is needed. In the literature a lot of studies on extreme sea level have long discussion whether a GEV or Pareto distribution should be used and whether joined variability issues should be treated and which peak threshold are to be used and what declustering of the data. This is relevant to discuss as for many applications the GEV is maybe not the best strategy, so make the reader aware or better, but demanding expand the tool in this direction.*

I will expand the section with some discussion on how the simulator could be expanded to work also with GPD distributions. In fact, any distribution can be used in practice. I don't really have much to add to the discussion about best thresholds and declustering. Generally speaking I think these parameters

are site specific and that it is hard to generalize.

*Line 98 please rephrase"is goes"*

This will be fixed.

*Line 115. If I understand correctly the GEV itself is not changing over time due to the climate change itself. Probably the only thing you can do, but likely to be incorrect as well. Most climate variables change in the mean and in the pdf. So you at least have to create awareness among your readers of this point.*

I will add a discussion about this. From a technical standpoint it would be easy to implement SSP based trends in the yearly sea level maximum as a part of the mean sea level projection. This requires no changes to the code at all. The problem, of course, is that suitable values for such trends are almost never known. I recently looked at trends in yearly sea level maximum in an ensemble of downscaled CMIP5 projections at a number of Swedish tide gauge locations. In that ensemble, natural variability was so much larger than any possible emission driven trend that detection was not possible. I expect this to be the case at many locations.

*Line 141 you have to mention that SSP scenarios don't have a probability in their definition, but you implicitly use a probability that they are equally likely if I understand correctly, this is not how they are defined*

I will make sure to mention that they have not been given probabilities by their makers in the revised version. However, I don't assume them to be equally likely. One of the key capabilities of the simulator is that the user can use his/her personal probabilities. The choice of probabilities is free and the simulator is an excellent tool to determine how changing these probabilities affect the flood risk. Hieronymus (2021) investigates this in more detail and I will add a more detailed explanation in the revised manuscript.

*Line 164 rephrase ones?*
I will rephrase.

*Line 195 That seems a trivial discussion. It is always correct in the extremes. At t=0 the trend is zero so extremes rule. If t goes to infinity SLR goes to extreme large values at least for high scenarios and the trend will rule. Explain this better in forehand and than show your examples where results are a mixture and both components might be important.*

I agree that this is a trivial point in the asymptotic sense of comparing $t = 0$ to $t = \infty$. However, where the simulator is helpful is over much shorter planning periods than that ending in $t = \infty$. I would argue that it is not trivial to determine when the flood risk transitions from being extreme to mean

sea level dominated. Clearly, this is both a location and scenario probability dependent problem. I will also argue that this type of knowledge is useful and could be used to improve coastal spatial planning so I think it is an important point to push.

---

## Author Response (AR1)

**Response to the questions of reviewer one**

First of all I would like thank the referee for his/her thoughtful comments, time and interest in the manuscript. I have put the referees comments in italics and my answers are given in plain text. Moreover, apart from correcting errors and following suggestions the referees have found. I also found an error of my own. I had consistently called the AR6 (low confidence) projections (low probability) projections. This error is now also fixed.

*Hieronymus presents the sea level simulator v1.0 following the first official public release of the source code. The sea level simulator aims at providing the joint probability of mean sea level rise and sea level extremes which is crucial for coastal planning and future adaptation. This tool could be particularly useful for engineering purpose and decision-making.*

*The manuscript acts as a technical reference publication as stated by the author. The framework behind the sea level simulator is not new and studies using previous versions of it have previously been published (Hieronymus, 2021; Hieronymus and Kalén, 2022). This version offers some updates such as new mean sea level projections (based on Fox-Kemper et al., 2021) and the implementation of uncertainty in the GEV parameters for the extreme sea levels.*

*The second part of the paper focus on a particular site and shows how the model behaves based on different parameter choices. More importantly, the manuscript discusses what one can infer from the model outputs. For example, which physical process is the dominant at different time horizons; this has direct impact when designing new structures and their planning periods.*

*The code is written in Matlab and released under MIT licence. Having no access to a Matlab licence, I have not tested the code directly, but the Gitlab is easily accessible. Users could potentially adapt and change the routines thanks to the use of Matlab. The different functions seem relatively straightforward to read though some users might find difficult to follow due to the lack of comments and descriptions inside each Matlab file.*

*GMD seems to be a good fit for the manuscript considering the author aim to publish a reference technical publication on the sea level simulator. Considering the previous two peer-reviewed papers on the framework, the comments below are mostly minor but would make the paper easier to digest by readers.*

*General Comments:*

*A significant part of the paper looks at what one can infer from the simulator (is the extreme level or SLR level the key driver, at which time horizon, etc), but it's not mentioned in the abstract. I think it is a very interesting capability and a key aspect of the simulator as it will be part of the decision-process a user*

*might look at. It should be a bit more highlighted.*

A new sentence has been added to the abstract and a former one has been slightly extended, see L6-7.

*As this paper should be viewed as a technical documentation, I think it would highly benefit from an extended Figure 2 where each step is associated with a sub-figure showing the raw time serie data, the GEV fits for extreme levels, the SLR scenarios, the sampling etc. It would really guide the reader. (I would also use more the word components / modules instead of nodes).*

A new Fig 2. has been made with subfigures for many of the steps, and I now use modules instead of nodes.

*"An example probability range is shown in Tab. 1" line 138; it might be good to have some guidelines for a user in how to choose these probabilities. As shown in the second part of the paper, they do have an impact.*

A discussion and some references about how one can think about these probabilities have been added on L155. Some discussion also exist in the last paragraph of the manuscript.

*I wonder if having another site with a potentially different behaviours or with different probability for the scenarios could be useful for a reader.*

This is the only reviewer suggestion that I object to. The primary reason for this is that I believe it would be very hard to keep the article as a technical reference publication. I think it would almost certainly drift towards discussing interesting questions about how regionally varying oceanographic conditions affect flood risk. This is, of course, an interesting topic, but one that in my opinion deserves its own article. A secondary reason is that in the Swedish context this has already been done in Hieronymus & Kalén (2022), where six sites with very long observational records were simulated.

*More extensive comments in the source would be beneficial to the users (mainly in emulator_ringhals_par.m, example_plot.m which seems to be the drivers); the associated set of functions are much more commented.*

Scripts updated with additional comments are now available in https://github.com/m-hieronymus/the_sea_level_simulator.

*The figures could be improved for better readability: none have a grid, the labels, ticks, and legends have generally small font size. I also believe showing the y-axis in term of return period would be more intuitive to a user / reader. This could be done on the right side with a twin axis for an example. The return period is stated in the text but shown in figure would be appreciated: "For*

*example, a frequency of 104 means that one in ten thousand planning periods contain a sea level of this height."*

All figures have been redrawn with grids, larger fonts, etc. Regarding the double axis for return period I did decide to go another way and just use the term planning period probability everywhere. A justification for this is given on L200. I believe that using the term planning period probability consistently is a good way of eliminating the confusion that always exist with return periods that change in time as a function of mean sea level change. I considered using a planning period period as the inverse of the planning period probability, but apart from having a period heavy name, I think it would also be more confusing than helpful given that the period in question would have to be defined in a cyclic time.

*Specific Comments:*

*Line 80: "very close to the ideal"; does the model keep its scaling skills for 16, 32 cores?*

My test computer only has four cores, but the answer is nonetheless almost certainly yes. That is, as long as you make a large amount of simulations so that each worker can get a significant workload. As mentioned in the manuscript the problem of running these simulations is, in fact, embarrassingly parallel so simulations can be run simultaneously on different computers and averaged afterwards. The overhead related to initialization, loading data and such is really small when millions of simulations are run.

*Figure 2 caption: "is goes"; remove "is"*

done.

*Line 100: "that reliably captures high frequency sea level variability"; maybe precise for the reader what "high-frequency" means so they do plug the right data to the model.*

A sentence is added on L98.

*Line 127-128: "These two scenarios have also low confidence projections, where the contribution from the Greenland and Antarctic ice-sheets to sea level rise is taken from some of the highest projections in the published scientific literature." A citation would be helpful here.*

Citations are added on L144.

*Line 145-147: I find this sentence not very clear.*

It is rewritten, see L166.

*Line 157: it could be useful to precise which other distributions have been tested to know what does and doesn't work.*

I made a note on L178.

*Line 161: "This script is easily edited for use with different mean sea level projections"; can easily be edited ?*

fixed.

*Line 179: "For example, a frequency of $10^{-4}$ mean that one in ten thousand planning periods contain a sea level of this height." Replace mean by means*

fixed.

*Line 183: maybe more "sea" than "ocean".*

fixed.

*Line 186: It would be good to reference here some previous studies when discussing the stratification in the North Sea / Baltic Sea regions. A recent study I found that could help finding the most appropriate references https://esd.copernicus.org/articles/13/373/2022/*

References are added on L221.

*Figure 8: what is the colorbar for? or is it lost in the middle of the dark dots.*

I forgot to turn on the shading. The figure is redrawn.

*Figure 8: I might misread the figure but why the extreme levels impact reduces with longer sea level maximum for different length of planning periods? Why not constant.*

You are reading it correctly. The reason is simply that when the planning period is short the joint sea level (i.e. mean+extreme) is almost equal to the extreme contribution. The mean sea level simply does not change that much in any projection between 2020 and 2050 and the only way to get say a 3 m joint sea level event is to have an extreme sea level of almost 3 m (a very unlikely extreme at Ringhals). Conversely, when your planning period is very long, mean sea level change can be sizeable in many different projections and percentiles. Therefore, there are many more combinations of mean and extreme in the 2020-2150 planning period that gives a 3 m joint sea level that contain a large mean

sea level contribution and a more average extreme contribution than the opposite.

**Response to the questions of reviewer two**

First of all I would like thank the referee for his/her thoughtful comments, time and interest in the manuscript. I have put the referees comments in italics and my answers are given in plain text. Moreover, apart from correcting errors and following suggestions the referees have found. I also found an error of my own. I had consistently called the AR6 (low confidence) projections (low probability) projections. This error is now also fixed.

*Review Hieronymus. The sea level simulator*

*This paper describes a simple statistical tool that combines trends in mean sea level and extremes for planning purposes. The paper is reasonably easy to access but requires here and there a bit more context to ensure proper use of the simulator. Below some suggestions for change which are all easy to accommodate in my view.*

*Line 73. A more conceptual introduction to the simulator is needed. You can not expect the reader to be familiar with Hieronymous 2021 and Hieronymous 2023.*

More conceptual introduction has been added to the paragraph starting on L40 with additional info in the paragraph starting on L57.

*Line 75. From the blue sky you start to discuss the parallelization aspects of the simulator. It is good to mention them somewhere in the paper, but not right at the start. In the end, being used to using large models, I don't think the parallelization is critical for the users to decide whether they use the tool or not. So please move backward in the paper or to an appendix.*

The paralellization part has been moved to a subsection at the end of Sect. 2.

*Line 86 define what a planning period is.*

A definition is given on L44.

*Line 95 Explain in more detail how mean sea level is combined with extreme information this can be done in various ways. A paragraph of discussion is needed. In the literature a lot of studies on extreme sea level have long discussion whether a GEV or Pareto distribution should be used and whether joined variability issues should be treated and which peak threshold are to be used and what declustering of the data. This is relevant to discuss as for many applications the GEV is maybe not the best strategy, so make the reader aware or better, but demanding expand the tool in this direction.*

A paragraph is added on L121. It does not go as as far as to implement this

functionality, but explains how it could be done.

*Line 98 please rephrase"is goes"*

fixed.

*Line 115. If I understand correctly the GEV itself is not changing over time due to the climate change itself. Probably the only thing you can do, but likely to be incorrect as well. Most climate variables change in the mean and in the pdf. So you at least have to create awareness among your readers of this point.*

A discussion about, and method for including trends in yearly sea level maximum now is discussion on L131

*Line 141 you have to mention that SSP scenarios don't have a probability in their definition, but you implicitly use a probability that they are equally likely if I understand correctly, this is not how they are defined*

The user is free to choose these probabilities him/herself so they are not necessarily equal (L5). Additional info about scenario probabilities is added on L155.

*Line 164 rephrase ones?*

fixed.

*Line 195 That seems a trivial discussion. It is always correct in the extremes. At t=0 the trend is zero so extremes rule. If t goes to infinity SLR goes to extreme large values at least for high scenarios and the trend will rule. Explain this better in forehand and than show your examples where results are a mixture and both components might be important.*

I agree that this is a trivial point in the asymptotic sense of comparing $t = 0$ to $t = \infty$. However, where the simulator is helpful is over much shorter planning periods than that ending in $t = \infty$. I would argue that it is not trivial to determine when the flood risk transitions from being extreme to mean sea level dominated. Clearly, this is both a location and scenario probability dependent problem. I will also argue that this type of knowledge is useful and could be used to improve coastal spatial planning so I think it is an important point to push. I have now extended the abstract to also reflect this capability of the simulator.

---

## Referee Report (RR1)

This new version of the manuscript has been improved from the original one and the author has considered most of my comments. I still have a couple of items I would like to see in the final version. Otherwise, the list of minor comments can be found below.

General Comments:

- The author discussed in the method section (lines 131-136) about the fact that change in extremes is not accounted for i.e. by 2150, the model considers extremes will follow the same statistics as present time. I think it's an important point and the author point out if such data were existing, they could be plugged into the model.

    I think it would be good to restate this point / discuss it in the conclusion as a caveat.

- I still find the figure font quite small, and I still believe having a twin axis with return period in addition of frequencies would be useful for a reader. Especially as the tool is intended for decision makers; a return period is more likely something perceived by the potential users compared to a frequency.

Minor Comments:

- line 36: "This is of course a great hindrance", maybe "barrier" or "Obstacle" instead of "hindrance"; this is just a suggestion
- Line 66-68: "the first paper…" and later "second paper"; Citations are missing. Which papers? also I would suggest using "study" or "work" instead of paper
- Line 70: maybe itemise the new features?
- There are a multitude of figure reference with the ")" I would remove it. E.g. instead of "Fig. 1b)" just write "Fig. 1b" etc – all the references to figure 1 basically.
- In Figure 1, the location of b) is not great. Top left corner would be better.
- Line 108. This sentence is not needed. This has already been explained few times. "Each such one armed bandit implies that there is a random process in operation when going from one module to the next."
- Around line 150, maybe add a number to each module in figure 1 so you can refer it in text. The sentence would be more fluid. For example: "in the third and forth module from the left in Fig. 1 " could be simplified to "in the third and **fourth** module**S**"
- Line 156: "ascribed" maybe "attributed" instead – again just a suggestion.
- Line 233: "it is plain to see that both the mean and the mean + extreme panel of the plot are dominated by this scenario at frequencies lower than approximately $10^{-3}$." not fully sure it is that obvious without comparing to the simulation without the low confidence scenarios.
- Line 281-282: The upper panels show the relative density of the mean and extreme sea level contributions to the joint sea level maxima. Reverse to match how they appear in figure (left is extreme and right mean).
- Line 286: "does give" maybe just "give"
- Line 308: "examples by (Hieronymus, 2021; Hieronymus and Kalén, 2022)." Remove parenthesis.
- Line 314: "The list of possible new applications is very long." replace "very long" by "extensive"
- Line 320: "find the the inverse" remove one "the"

---

## Author Response (AR2)

**Response to the questions of reviewer one**

First of all I would like to again thank the referee for his/her thoughtful comments, time and interest in the manuscript. I have put the referees comments in italics and my answers are given in plain text.

*This new version of the manuscript has been improved from the original one and the author has considered most of my comments. I still have a couple of items I would like to see in the final version. Otherwise, the list of minor comments can be found below.*

*General Comments:*

*The author discussed in the method section (lines 131-136) about the fact that change in extremes is not accounted for i.e. by 2150, the model considers extremes will follow the same statistics as present time. I think it's an important point and the author point out if such data were existing, they could be plugged into the model.*

*I think it would be good to restate this point / discuss it in the conclusion as a caveat.*

A discussion has been added on L319-324

*I still find the figure font quite small, and I still believe having a twin axis with return period in addition of frequencies would be useful for a reader. Especially as the tool is intended for decision makers; a return period is more likely something perceived by the potential users compared to a frequency.*

In this update I have redrawn all figures using practically the largest font I could fit within the panels.

Regarding the twin y-axes, I was unclear in the first response. It is not just a preference of mine to use planning period probability instead of yearly probability or both. It is, in fact impossible to translate planning period probability into a unique return period in these plots. In other words, these plots cannot be made with a twin y-axis showing yearly probability (1/ return period) and planning period probability for three different planning periods simultaneously. I will try to give a more detailed explanation in the following.

Firstly, the CDFs from which the return period (1/yearly probability) could be determined for the mean and mean + extreme panels are time dependent. That is, the return period of seeing say a 2 m sea level today at Ringhals is perhaps 10000 years, while the same sea level might have a return period of perhaps 5 years in 2150. The planning period probability is fixed during a planning period, while the yearly probability is not. Therefore there is no one to one

correspondence between planning period probability and yearly probability, so a single extra y-axis cannot do the translation.

Secondly, even for the extreme only panel, which has a stationary distribution there is no one to one correspondence between planning period probability and yearly probability. In fact, in this case, it is the return level i.e. sea level above the mean that has a one to one correspondence with return period (like in Fig. 1b). That, is it would be more natural with double x-axes than double y-axes in this case. This can also be seen by noting that for every x-value in the extreme only panel there are three y-values, one for each planning period length. Therefore, one would need three different y-axes or a double x-axis in this case. However, the double x-axis would be very hard to fit, and it would not work in the other panels because of the non-stationary distributions.

In conclusion, I agree that it would be informative for planners if planning period probabilities could be translated into yearly probabilities, but I don't believe it is possible in those figures. In Hieronymus & Kalén (2022) we did calculate the yearly probability of the extreme component of joint (mean+extreme) sea level maximum with a fixed 1/10000 planning period probability. This I think is a useful diagnostic, but it relies on first freezing the planning period probability, so it would not work in the figures shown here. Ultimately, I believe that thinking in terms of return period is only useful for processes that are stationary of close to stationary, so for sea level I would advocate moving away from them.

*Minor Comments:*

*line 36: "This is of course a great hindrance", maybe "barrier" or "Obstacle" instead of "hindrance"; this is just a suggestion*

I have now changed to obstacle.

*Line 66-68: "the first paper..." and later "second paper"; Citations are missing. Which papers? also I would suggest using "study" or "work" instead of paper*

Citations are added and the word paper is changed to study.

*Line 70: maybe itemise the new features?*

They are now itemized.

*There are a multitude of figure reference with the ")" I would remove it. E.g. instead of "Fig.1b)" just write "Fig. 1b" etc – all the references to figure 1 basically.*

I have redone the referencing according to the reviewers suggestion.

*In Figure 1, the location of b) is not great. Top left corner would be better.*

The b) is moved to the top left corner.

*Line 108. This sentence is not needed. This has already been explained few times. "Each such one armed bandit implies that there is a random process in operation when going from one module to the next."*

It is now removed.

*Around line 150, maybe add a number to each module in figure 1 so you can refer it in text. The sentence would be more fluid. For example: "in the third and forth module from the left in Fig.1 " could be simplified to "in the third and fourth moduleS"*

Numbers have been added to the modules in Fig. 1a and the sentence has been rewritten according to the reviewers suggestion.

*Line 156: "ascribed" maybe "attributed" instead – again just a suggestion.*

I changed to attributed.

*Line 233: "it is plain to see that both the mean and the mean + extreme panel of the plot are dominated by this scenario at frequencies lower than approximately 10 3 ." not fully sure it is that obvious without comparing to the simulation without the low confidence scenarios.*

The plain to see part has been removed.

*Line 281-282: The upper panels show the relative density of the mean and extreme sea level contributions to the joint sea level maxima. Reverse to match how they appear in figure (left is extreme and right mean).*

I have switched extreme and mean.

*Line 286: "does give" maybe just "give"*

I switched to just give.

*Line 308: "examples by (Hieronymus, 2021; Hieronymus and Kalén, 2022)." Remove parenthesis.*

It have been removed.

*Line 314: "The list of possible new applications is very long." replace "very long" by "extensive"*

it now says extensive.

*Line 320: "find the the inverse" remove one "the"*

one the has been removed.